# Showing versus Doing: Teaching by Demonstration

**Mark K Ho**
Department of Cognitive, Linguistic, and Psychological Sciences
Brown University
Providence, RI 02912
mark_ho@brown.edu

**Michael L. Littman**
Department of Computer Science
Brown University
Providence, RI 02912
mlittman@cs.brown.edu

**James MacGlashan**
Department of Computer Science
Brown University
Providence, RI 02912
james_macglashan@brown.edu

**Fiery Cushman**
Department of Psychology
Harvard University
Cambridge, MA 02138
cushman@fas.harvard.edu

**Joseph L. Austerweil**
Department of Psychology
University of Wisconsin-Madison
Madison, WI 53706
austerweil@wisc.edu

## Abstract

People often learn from others' demonstrations, and inverse reinforcement learning (IRL) techniques have realized this capacity in machines. In contrast, *teaching* by demonstration has been less well studied computationally. Here, we develop a Bayesian model for teaching by demonstration. Stark differences arise when demonstrators are intentionally teaching (i.e. showing) a task versus simply performing (i.e. doing) a task. In two experiments, we show that human participants modify their teaching behavior consistent with the predictions of our model. Further, we show that even standard IRL algorithms benefit when learning from showing versus doing.

## 1   Introduction

Is there a difference between *doing* something and *showing* someone else how to do something? Consider cooking a chicken. To cook one for dinner, you would do it in the most efficient way possible while avoiding contaminating other foods. But, what if you wanted to teach a completely naïve observer how to prepare poultry? In that case, you might take pains to emphasize certain aspects of the process. For example, by ensuring the observer sees you wash your hands thoroughly after handling the uncooked chicken, you signal that it is undesirable (and perhaps even dangerous) for other ingredients to come in contact with raw meat. More broadly, how could an agent *show* another agent how to do a task, and, in doing so, teach about its underlying reward structure?

To model showing, we draw on psychological research on learning and teaching concepts by example. People are good at this. For instance, when a teacher signals their pedagogical intentions, children more frequently imitate actions and learn abstract functional representations [6, 7]. Recent work has formalized concept teaching as a form of recursive social inference, where a teacher chooses an example that best conveys a concept to a learner, who assumes that the teacher is choosing in this manner [14]. The key insight from these models is that helpful teachers do not merely select

probable examples of a concept, but rather choose examples that *best disambiguate* a concept from other candidate concepts. This approach allows for more effective, and more efficient, teaching and learning of concepts from examples.

We can extend these ideas to explain showing behavior. Although recent work has examined user-assisted teaching [8], identified legible motor behavior in human-machine coordination [9], and analyzed reward coordination in game theoretic terms [11], previous work has yet to successfully model how people naturally teach reward functions by demonstration. Moreover, in Inverse Reinforcement Learning (IRL), in which an observer attempts to infer the reward function that an expert (human or artificial) is maximizing, it is typically assumed that experts are only doing the task and not intentionally showing how to do the task. This raises two related questions: First, how does a person showing how to do a task differ from them just doing it? And second, are standard IRL algorithms able to benefit from human attempts to show how to do a task?

In this paper, we investigate these questions. To do so, we formulate a computational model of showing that applies Bayesian models of teaching by example to the reward function learning setting. We contrast this *pedagogical model* with a model of doing: standard optimal planning in Markov Decision Processes. The pedagogical model predicts several systematic differences from the standard planning model, and we test whether human participants reproduce these distinctive patterns. For instance, the pedagogical model chooses paths to a goal that best disambiguates which goal is being pursued (Experiment 1). Similarly, when teaching feature-based reward functions, the model will prioritize trajectories that better signal the reward value of state features or even perform trajectories that would be inefficient for an agent simply doing the task (Experiment 2). Finally, to determine whether showing is indeed better than doing, we train a standard IRL algorithm with our model trajectories and human trajectories.

## 2 A Bayesian Model of Teaching by Demonstration

Our model draws on two approaches: IRL [2] and Bayesian models of teaching by example [14]. The first of these, IRL and the related concept of inverse planning, have been used to model people's theory of mind, or the capacity to infer another agent's unobservable beliefs and/or desires through their observed behavior [5]. The second, Bayesian models of pedagogy, prescribe how a teacher should use examples to communicate a concept to an ideal learner. Our model of teaching by demonstration, called Pedagogical Inverse Reinforcement Learning, merges these two approaches together by treating a teacher's demonstration trajectories as communicative acts that signal the reward function that an observer should learn.

### 2.1 Learning from an Expert's Actions

#### 2.1.1 Markov Decision Processes

An agent that plans to maximize a reward function can be modeled as the solution to a Markov Decision Process (MDP). An MDP is defined by the tuple $< \mathcal{S}, \mathcal{A}, T, R, \gamma >$: a set of states in the world $\mathcal{S}$; a set of actions for each state $\mathcal{A}(s)$; a transition function that maps states and actions to next states, $T : \mathcal{S} \times \mathcal{A} \to \mathcal{S}$ (in this work we assume all transitions are deterministic, but this can be generalized to probabilistic transitions); a reward function that maps states to scalar rewards, $R : \mathcal{S} \to \mathbb{R}$; and a discount factor $\gamma \in [0, 1]$. Solutions to an MDP are stochastic policies that map states to distributions over actions, $\pi : \mathcal{S} \to P(\mathcal{A}(s))$. Given a policy, we define the expected cumulative discounted reward, or *value*, $V^\pi(s)$, at each state associated with following that policy:

$$V^\pi(s) = E_\pi \Big[ \sum_{k=0}^{\infty} \gamma^k r_{t+k+1} \mid s_t = s \Big]. \tag{1}$$

In particular, the optimal policy for an MDP yields the *optimal value function*, $V^*$, which is the value function that has the maximal value for every state ($V^*(s) = \max_\pi V^\pi(s), \forall s \in \mathcal{S}$). The optimal policy also defines an optimal state-action value function, $Q^*(s, a) = E_\pi[r_{t+1} + \gamma V^*(s_{t+1}) \mid s_t = s, a_t = a]$.

---

**Algorithm 1** Pedagogical Trajectory Algorithm

---

**Require:** starting states $\mathbf{s}$, reward functions $\{R_1, R_2, ..., R_N\}$, transition function $T$, maximum showing trajectory depth $l_{max}$, minimum hypothetical doing probability $p_{min}$, teacher maximization parameter $\alpha$, discount factor $\gamma$.

1: $\mathbf{\Pi} \leftarrow \emptyset$
2: **for** $i = 1$ to $N$ **do**
3:     $Q_i = \text{calculateActionValues}(\mathbf{s}, R_i, T, \gamma)$
4:     $\pi_i = \text{softmax}(Q_i, \lambda)$
5:     $\mathbf{\Pi}.\text{add}(\pi_i)$
6: Calculate $\mathbf{j} = \{j : s_1 \in \mathbf{s}, \; length(j) \leq l_{max}, \text{ and } \exists \pi \in \mathbf{\Pi} \text{ s.t. } \prod_{(s_i,a_i) \in j} \pi(a_i \mid s_i) > p_{min}\}$.
7: Construct hypothetical doing probability distribution $P_{\text{Doing}}(j \mid R)$ as an N x M array.
8: $P_{\text{Observing}}(R \mid j) = \frac{P_{\text{Doing}}(j|R)P(R)}{\sum_{R'} P_{\text{Doing}}(j|R')P(R')}$
9: $P_{\text{Showing}}(j \mid R) = \frac{P_{\text{Observing}}(R|j)^\alpha}{\sum_{j'} P_{\text{Observing}}(R|j')^\alpha}$
10: **return** $P_{\text{Showing}}(j \mid R)$

---

### 2.1.2 Inverse Reinforcement Learning (IRL)

In the Reinforcement Learning setting, an agent takes actions in an MDP and receives rewards, which allow it to eventually learn the optimal policy [15]. We thus assume that an expert who knows the reward function and is *doing* a task selects an action $a_t$ in a state $s_t$ according to a Boltzmann policy, which is a standard soft-maximization of the action-values:

$$P_{\text{Doing}}(a_t \mid s_t, R) = \frac{\exp\{Q^*(s_i, a_i)/\lambda\}}{\sum_{a' \in \mathcal{A}(s_i)} \exp\{Q^*(s_i, a')/\lambda\}}. \tag{2}$$

$\lambda > 0$ is an inverse temperature parameter (as $\lambda \to 0$, the expert selects the optimal action with probability 1; as $\lambda \to \infty$, the expert selects actions uniformly randomly).

In the IRL setting, an observer sees a trajectory of an expert executing an optimal policy, $j = \{(s_1, a_1), (s_2, a_2), ..., (s_k, a_k)\}$, and infers the reward function $R$ that the expert is maximizing. Given that an agent's policy is stationary and Markovian, the probability of the trajectory given a reward function is just the product of the individual action probabilities, $P_{\text{Doing}}(j \mid R) = \prod_t P_{\text{Doing}}(a_t \mid s_t, R)$. From a Bayesian perspective [13], the observer is computing a posterior probability over possible reward functions $\mathcal{R}$:

$$P_{\text{Observing}}(R \mid j) = \frac{P_{\text{Doing}}(j \mid R)P(R)}{\sum_{R'} P_{\text{Doing}}(j \mid R')P(R')}. \tag{3}$$

Here, we always assume that $P(R)$ is uniform.

### 2.2 Bayesian Pedagogy

IRL typically assumes that the demonstrator is executing the stochastic optimal policy for a reward function. But is this the best way to teach a reward function? Bayesian models of pedagogy and communicative intent have shown that choosing an example to teach a concept differs from simply sampling from that concept [14, 10]. These models all treat the teacher's choice of a datum, $d$, as maximizing the probability a learner will infer a target concept, $h$:

$$P_{\text{Teacher}}(d \mid h) = \frac{P_{\text{Learner}}(h \mid d)^\alpha}{\sum_{d'} P_{\text{Learner}}(h \mid d')^\alpha}. \tag{4}$$

$\alpha$ is the teacher's softmax parameter. As $\alpha \to 0$, the teacher chooses uniformly randomly; as $\alpha \to \infty$, the teacher chooses $d$ that maximally causes the learner to infer a target concept $h$; when $\alpha = 1$, the teacher is "probability matching".

The teaching distribution describes how examples can be effectively chosen to teach a concept. For instance, consider teaching the concept of "even numbers". The sets $\{2, 2, 2\}$ and $\{2, 18, 202\}$ are both examples of even numbers. Indeed, given finite options with replacement, they both have the same probability of being randomly chosen as sets of examples. But $\{2, 18, 202\}$ is clearly better

for helpful teaching since a naïve learner shown $\{2, 2, 2\}$ would probably infer that "even numbers" means "the number 2". This illustrates an important aspect of successful teaching by example: that examples should not only be consistent with the concept being taught, but should also *maximally disambiguate* the concept being taught from other possible concepts.

## 2.3 Pedagogical Inverse Reinforcement Learning

To define a model of teaching by demonstration, we treat the teacher's trajectories in a reinforcement-learning problem as a "communicative act" for the learner's benefit. Thus, an effective teacher will modify its demonstrations when *showing* and not simply *doing* a task. As in Equation 4, we can define a teacher that selects trajectories that best convey the reward function:

$$P_{\text{Showing}}(j \mid R) = \frac{P_{\text{Observing}}(R \mid j)^{\alpha}}{\sum_{j'} P_{\text{Observing}}(R \mid j')^{\alpha}}. \tag{5}$$

In other words, *showing* depends on a demonstrator's inferences about an observer's inferences about *doing*.

This model provides quantitative and qualitative predictions for how agents will show and teach how to do a task given they know its true reward function. Since humans are the paradigm teachers and a potential source of expert knowledge for artificial agents, we tested how well our model describes human teaching. In Experiment 1, we had people teach simple goal-based reward functions in a discrete MDP. Even though in these cases entering a goal is already highly diagnostic, different paths of different lengths are better for *showing*, which is reflected in human behavior. In Experiment 2, people taught more complex feature-based reward functions by demonstration. In both studies, people's behavior matched the qualitative predictions of our models.

# 3 Experiment 1: Teaching Goal-based Reward Functions

Consider a grid with three possible terminal goals as shown in Figure 1. If an agent's goal is $\&$, it could take a number of routes. For instance, it could move all the way right and then move upwards towards the $\&$ (right-then-up) or first move upwards and then towards the right (up-then-right). But, what if the agent is not just *doing* the task, but also attempting to *show* it to an observer trying to learn the goal location?

When the goal is $\&$, our pedagogical model predicts that up-then-right is the more probable trajectory because it is more disambiguating. Up-then-right better indicates that the intended goal is $\&$ than right-then-up because right-then-up has more actions consistent with the goal being $\#$. We have included an analytic proof of why this is the case for a simpler setting in the supplementary materials. Additionally, our pedagogical model makes the prediction that when trajectory length costs are negligible, agents will engage in repetitive, inefficient behaviors that gesture towards one goal location over others. This "looping" behavior results when an agent can return to a state with an action that has high signaling value by taking actions that have a low signaling "cost" (i.e. they do not signal something other than the true goal). Figure 1d shows an example of such a looping trajectory.

In Experiment 1, we tested whether people's showing behavior reflected the pedagogical model when reward functions are goal-based. If so, this would indicate that people choose the disambiguating path to a goal when showing.

## 3.1 Experimental Design

Sixty Amazon Mechanical Turk participants performed the task in Figure 1. One was excluded due to missing data. All participants completed a learning block in which they had to find the reward location without being told. Afterwards, they were either placed in a *Do* condition or a *Show* condition. Participants in *Do* were told they would win a bonus based on the number of rewards (correct goals) they reached and were shown the text, "The reward is at location X", where X was one of the three symbols $\%$, $\#$, or $\&$. Those in *Show* were told they would win a bonus based on how well a randomly matched partner who was shown their responses (and did not know the location of the reward) did on the task. On each round of *Show*, participants were shown text saying "Show your partner that the reward is at location X". All participants were given the same sequence of trials in which the reward locations were $<\%, \&, \#, \&, \%, \#, \%, \#, \&>$.

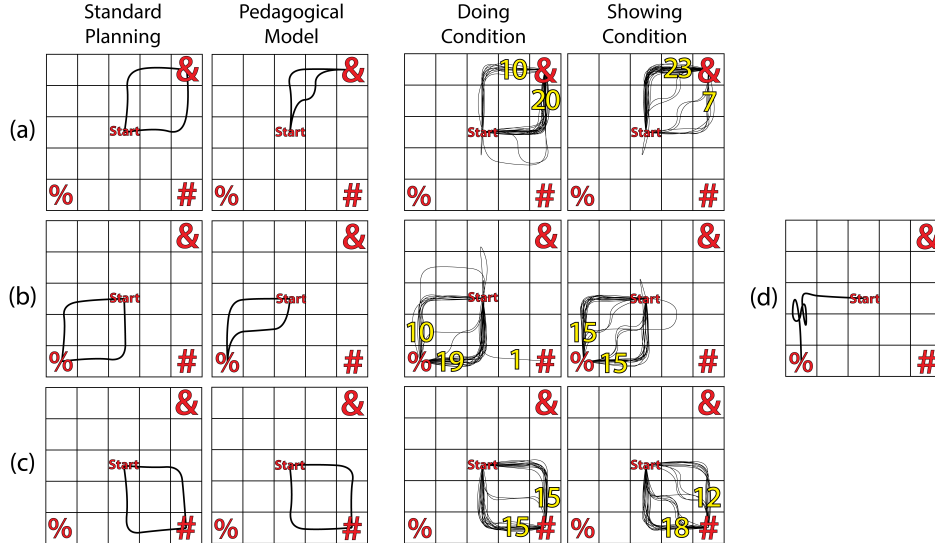

Figure 1: Experiment 1: Model predictions and participant trajectories for 3 trials when the goal is (a) &, (b) %, and (c) #. Model trajectories are the two with the highest probability ($\lambda = 2$, $\alpha = 1.0$, $p_{min} = 10^{-6}$, $l_{max} = 4$). Yellow numbers are counts of trajectories with the labeled tile as the penultimate state. (d) An example of looping behavior predicted by the model when % is the goal.

## 3.2 Results

As predicted, *Show* participants tended to choose paths that disambiguated their goal as compared to *Do* participants. We coded the number of responses on & and % trials that were "showing" trajectories based on how they entered the goal (i.e. out of 3 for each goal). On & trials, entering from the left, and on % trials, entering from above were coded as "showing". We ran a 2x2 ANOVA with *Show* vs *Do* as a between-subjects factor and goal (% vs &) as a repeated measure. There was a main effect of condition ($F(1, 57) = 16.17, p < .001$; *Show*: M = 1.82, S.E. 0.17; *Do*: M = 1.05, S.E. 0.17) as well as a main effect of goal ($F(1, 57) = 4.77, p < .05$; %-goal: M = 1.73, S.E. = 0.18; &-goal: M = 1.15, S.E. = 0.16). There was no interaction ($F(1, 57) = 0.98, p = 0.32$).

The model does not predict any difference between conditions for the # (lower right) goal. However, a visual analysis suggested that more participants took a "swerving" path to reach #. This observation was confirmed by looking at trials where # was the goal and comparing the number of swerving trials, which was defined as making more than one change in direction (*Show*: M = 0.83, *Do*: M = 0.26; two-sided t-test: $t(44.2) = 2.18, p = 0.03$). Although not predicted by the model, participants may swerve to better signal their intention to move 'directly' towards the goal.

## 3.3 Discussion

Reaching a goal is sufficient to indicate its location, but participants still chose paths that better disambiguated their intended goal. Overall, these results indicate that people are sensitive to the distinction between doing and showing, consistent with our computational framework.

## 4 Experiment 2: Teaching Feature-based Reward Functions

Experiment 1 showed that people choose disambiguating plans even when entering the goal makes this seemingly unnecessary. However, one might expect richer showing behavior when teaching more complex reward functions. Thus, for Experiment 2, we developed a paradigm in which showing how to do a task, as opposed to merely doing a task, makes a difference for how well the underlying reward function is learned. In particular, we focused on teaching feature-based reward functions that allow an agent to generalize what it has learned in one situation to a new situation. People often use feature-based representations for generalization [3], and feature-based reward functions have been used extensively in reinforcement learning (e.g. [1]). We used a colored-tile grid task shown in

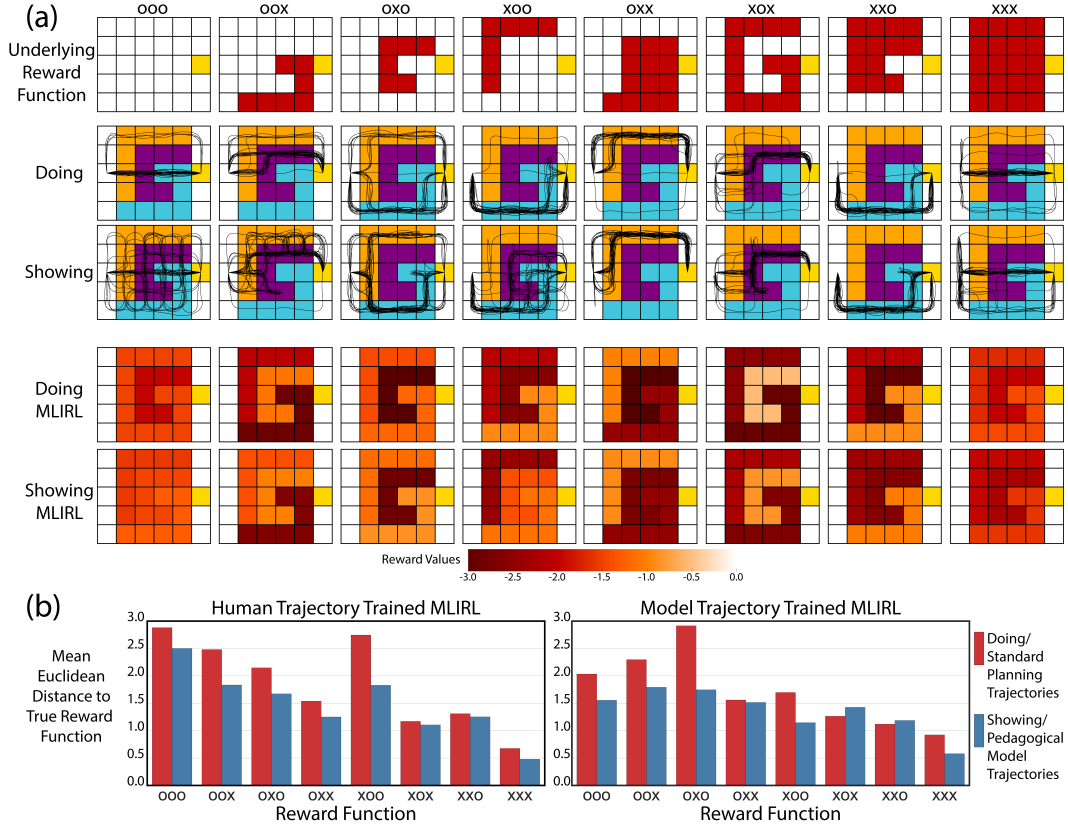

Figure 2: Experiment 2 results. (a) Column labels are reward function codes. They refer to which tiles were safe (o) and which were dangerous (x) with the ordering <orange, purple, cyan>. Row 1: Underlying reward functions that participants either did or showed; Row 2: *Do* participant trajectories with visible tile colors; Row 3: *Show* participant trajectories; Row 4: Mean reward function learned from *Do* trajectories by Maximum-Likelihood Inverse Reinforcement Learning (MLIRL) [4, 12]; Row 5: Mean reward function learned from *Show* trajectories by MLIRL. (b) Mean distance between learned and true reward function weights for human-trained and model-trained MLIRL. For the models, MLIRL results for the top two ranked demonstration trajectories are shown.

Figure 2 to study teaching feature-based reward functions. White tiles are always "safe" (reward of 0), while yellow tiles are always terminal states that reward 10 points. The remaining 3 tile types–orange, purple, and cyan–are each either "safe" or "dangerous" (reward of $-2$). The rewards associated with the three tile types are independent, and nothing about the tiles themselves signal that they are safe or dangerous.

A standard planning algorithm will reach the terminal state in the most efficient and optimal manner. Our pedagogical model, however, predicts that an agent who is showing the task will engage in specific behaviors that best disambiguate the true reward function. For instance, the pedagogical model is more likely to take a roundabout path that leads through all the safe tile types, choose to remain on a safe colored tile rather than go on the white tiles, or even loop repeatedly between multiple safe tile-types. All of these types of behaviors send strong signals to the learner about which tiles are safe as well as which tiles are dangerous.

## 4.1 Experimental Design

Sixty participants did a feature-based reward teaching task; two were excluded due to missing data. In the first phase, all participants were given a learning-applying task. In the learning rounds, they interacted with the grid shown in Figure 2 while receiving feedback on which tiles won or lost points.

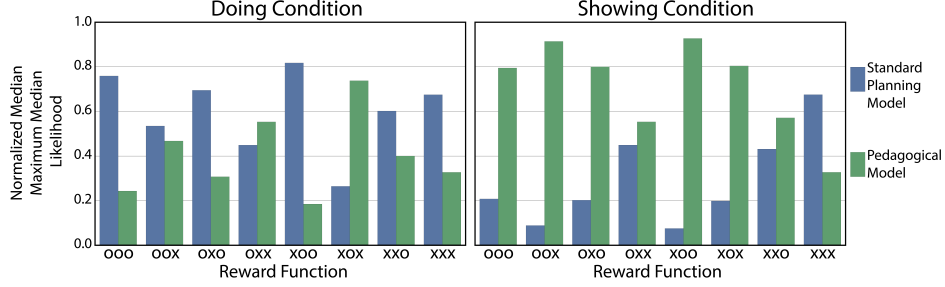

Figure 3: Experiment 2 normalized median model fits.

Safe tiles were worth 0 points, dangerous tiles were worth -2 points, and the terminal goal tile was worth 5 points. They also won an additional 5 points for each round completed for a total of 10 points. Each point was worth 2 cents of bonus. After each learning round, an applying round occurred in which they applied what they just learned about the tiles without receiving feedback in a new grid configuration. They all played 8 pairs of learning and applying rounds corresponding to the 8 possible assignments of "safe" and "dangerous" to the 3 tile types, and order was randomized between participants.

As in Experiment 1, participants were then split into *Do* or *Show* conditions with no feedback. *Do* participants were told which colors were safe and won points for performing the task. *Show* participants still won points and were told which types were safe. They were also told that their behavior would be shown to another person who would apply what they learned from watching the participant's behavior to a separate grid. The points won would be added to the demonstrator's bonus.

## 4.2 Results

Responses matched model predictions. *Do* participants simply took efficient routes, whereas *Show* participants took paths that signaled tile reward values. In particular, *Show* participants took paths that led through multiple safe tile types, remained on safe colored tiles when safe non-colored tiles were available, and looped at the boundaries of differently colored safe tiles.

### 4.2.1 Model-based Analysis

To determine how well the two models predicted human behaviors globally, we fit separate models for each reward function and condition combination. We found parameters that had the highest median likelihood out of the set of participant trajectories in a given reward function-condition combination. Since some participants used extremely large trajectories (e.g. >25 steps) and we wanted to include an analysis of all the data, we calculated best-fitting state-action policies. For the standard-planner, it is straightforward to calculate a Boltzmann policy for a reward function given $\lambda$.

For the pedagogical model, we first need to specify an initial model of doing and distribution over a finite set of trajectories. We determine this initial set of trajectories and their probabilities using three parameters: $\lambda$, the softmax parameter for a hypothetical "doing" agent that the model assumes the learner believes it is observing; $l_{max}$, the maximum trajectory length; and $p_{min}$, the minimum probability for a trajectory under the hypothetical doing agent. The pedagogical model then uses an $\alpha$ parameter that determines the degree to which the teacher is maximizing. State-action probabilities are calculated from a distribution over trajectories using the equation $P(a \mid s, R) = \sum_j P(a \mid s, j)P(j \mid R)$, where $P(a \mid s, j) = \frac{|\{(s,a):s=s_t, a=a_t \forall (s_t,a_t) \in j\}|}{|\{(s,a):s=s_t \forall (s_t,a_t) \in j\}|}$.

We fit parameter values that produced the maximum median likelihood for each model for each reward function and condition combination. These parameters are reported in the supplementary materials. The normalized median fit for each of these models is plotted in Figure 3. As shown in the figure, the standard planning model better captures behavior in the *Do* condition, while the pedagogical model better captures behavior in the *Show* condition. Importantly, even when the standard planning model could have a high $\lambda$ and behave more randomly, the pedagogical model better fits the *Show* condition. This indicates that showing is not simply random behavior.

#### 4.2.2 Behavioral Analyses

We additionally analyzed specific behavioral differences between the *Do* and *Show* conditions predicted by the models. When showing a task, people visit a greater variety of safe tiles, visit tile types that the learner has uncertainty about (i.e. the colored tiles), and more frequently revisit states or "loop" in a manner that leads to better signaling. We found that all three of these behaviors were more likely to occur in the *Show* condition than in the *Do* condition.

To measure the variety of tiles visited, we calculated the entropy of the frequency distribution over colored-tile visits by round by participant. Average entropy was higher for *Show* (*Show*: M = 0.50, SE = 0.03; *Do*: M = 0.39, SE = 0.03; two-sided t-test: $t(54.9) = -3.27, p < 0.01$). When analyzing time spent on colored as opposed to un-colored tiles, we calculated the proportion of visits to colored tiles after the first colored tile had been visited. Again, this measure was higher for *Show* (*Show*: M = 0.87, SE = 0.01; *Do*: M = 0.82, SE = 0.01; two-sided t-test: $t(55.6) = -3.14, p < .01$). Finally, we calculated the number of times states were revisited in the two conditions–an indicator of "looping"–and found that participants revisited states more in *Show* compared to *Do* (*Show*: M = 1.38, SE = 0.22; *Do*: M = 0.10, SE = 0.03; two-sided t-test: $t(28.3) = -2.82, p < .01$). There was no difference between conditions in the total rewards won (two-sided t-test: $t(46.2) = .026, p = 0.80$).

### 4.3 Teaching Maximum-Likelihood IRL

One reason to investigate showing is its potential for training artificial agents. Our pedagogical model makes assumptions about the learner, but it may be that pedagogical trajectories are better even for training off-the-shelf IRL algorithms. For instance, Maximum Likelihood IRL (MLIRL) is a state-of-the-art IRL algorithm for inferring feature-based reward functions [4, 12]. Importantly, unlike the discrete reward function space our showing model assumes, MLIRL estimates the maximum likelihood reward function over a space of continuous feature weights using gradient ascent.

To test this, we input human and model trajectories into MLIRL. We constrained non-goal feature weights to be non-positive. Overall, the algorithm was able to learn the true reward function better from showing than doing trajectories produced by either the models or participants (Figure 2).

#### 4.3.1 Discussion

When learning a feature-based reward function from demonstration, it matters if the demonstrator is showing or doing. In this experiment, we showed that our model of pedagogical reasoning over trajectories captures how people show how to do a task. When showing as opposed to simply doing, demonstrators are more likely to visit a variety of states to show that they are safe, stay on otherwise ambiguously safe tiles, and also engage in "looping" behavior to signal information about the tiles. Moreover, this type of teaching is even better at training standard IRL algorithms like MLIRL.

## 5 General Discussion

We have presented a model of showing as Bayesian teaching. Our model makes accurate quantitative and qualitative predictions about human showing behavior, as demonstrated in two experiments. Experiment 1 showed that people modify their behavior to signal information about goals, while Experiment 2 investigated how people teach feature-based reward functions. Finally, we showed that even standard IRL algorithms benefit from showing as opposed to merely doing.

This provides a basis for future study into intentional teaching by demonstration. Future research must explore showing in settings with even richer state features and whether more savvy observers can leverage a showing agent's pedagogical intent for even better learning.

#### Acknowledgments

MKH was supported by the NSF GRFP under Grant No. DGE-1058262. JLA and MLL were supported by DARPA SIMPLEX program Grant No. 14-46-FP-097. FC was supported by grant N00014-14-1-0800 from the Office of Naval Research.

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
