[Supplementary Material · NIPS_2016_Supplementary_Materials.pdf]

# Teaching by Demonstration Supplementary Materials

Mark K Ho, Michael L. Littman, James MacGlashan,
Fiery Cushman, Joseph L. Austerweil

September 2016

## Appendix 1: An Example with 2 Goals

Suppose we have a 3x2 gridworld with two possible terminal goals (X and Y) and a starting position as shown in Figure 1i. We assume no step costs and $\gamma = .99$. We restrict our analysis to trajectories of length 2 that terminate at a goal state. Thus there are 4 trajectories considered.

Figure 1: (i) Gridworld with 2 possible goal states (labeled X and Y) and a single starting state. (ii) All trajectories of length 2 that terminate at a goal state.

## Proof

The purpose of this proof is to show that certain trajectories have higher probability of being chosen by a demonstrator who is "showing" as opposed to "doing" a task, even when all trajectories enter a goal. The prior probability over goals is uniform.

The following inequalities for a goal $g \in G = \{X, Y\}$ given a trajectory $j \in J = \{x_{in}, x_{out}, y_{in}, y_{out}\}$ will hold when a softmax policy or $\epsilon$-greedy policy is used to calculate the standard planning distribution:

$$P_{\text{Doing}}(x_{out} \mid X) \geq P_{\text{Doing}}(x_{in} \mid X) > 0 \tag{1}$$

$$P_{\text{Doing}}(x_{in} \mid Y) > P_{\text{Doing}}(x_{out} \mid Y) > 0. \tag{2}$$

An observer watching a standard planner uses Bayes rule to infer the goal being pursued:

$$P_{\text{Observing}}(G = g \mid J = j) = \frac{P_{\text{Doing}}(J = j \mid G = g)}{\sum_{g'} P_{\text{Doing}}(J = j \mid G = g')}. \tag{3}$$

The inequalities in (1) and (2) entail the following inequality[1]:

$$\frac{P_{\text{Doing}}(x_{out} \mid X)}{P_{\text{Doing}}(x_{out} \mid X) + P_{\text{Doing}}(x_{out} \mid Y)} > \frac{P_{\text{Doing}}(x_{in} \mid X)}{P_{\text{Doing}}(x_{in} \mid X) + P_{\text{Doing}}(x_{in} \mid Y)}. \tag{4}$$

$$P_{\text{Observing}}(X \mid x_{out}) > P_{\text{Observing}}(X \mid x_{in}). \tag{5}$$

That is, observing $x_{out}$ provides better evidence that X is the goal than observing $x_{in}$. Since an agent that is showing an observer will choose as follows:

$$P_{\text{Showing}}(J = j \mid G = g) = \frac{P_{\text{Observing}}(G = g \mid J = j)^{\alpha}}{\sum_{j'} P_{\text{Observing}}(G = g \mid J = j')^{\alpha}}, \tag{6}$$

then,

$$\frac{P_{\text{Observing}}(X \mid x_{out})^{\alpha}}{\sum_{j'} P_{\text{Observing}}(X \mid j')^{\alpha}} > \frac{P_{\text{Observing}}(X \mid x_{in})^{\alpha}}{\sum_{j'} P_{\text{Observing}}(X \mid j')^{\alpha}} \tag{7}$$

$$P_{\text{Showing}}(x_{out} \mid X) > P_{\text{Showing}}(x_{in} \mid X) \tag{8}$$

Intuitively, the different probabilities of $x_{out}$ and $x_{in}$ when $Y$ is the goal allows a showing agent to "break the symmetry" between $x_{out}$ and $x_{in}$ when $X$ is the goal. Analogous calculations can show that $P_{\text{Showing}}(y_{out} \mid Y) > P_{\text{Showing}}(y_{in} \mid Y)$.

$$ac > bd$$
$$ab + ac > bd + ab$$
$$a(b + c) > b(a + d)$$
$$\frac{a}{a + d} > \frac{b}{b + c}$$

# Appendix 2: Experiment 2 Model Fits

Table 1: Experiment 2 Maximum Median Likelihood Model Parameters

| | | Doing Condition | | | | | | | |
|---|---|---|---|---|---|---|---|---|---|
| | | ooo | oox | oxo | oxx | xoo | xox | xxo | xxx |
| Standard Planning Model | $\lambda$ | 0.02 | 0.02 | 0.02 | 0.02 | 0.02 | 0.04 | 0.02 | 0.02 |
| Pedagogical Model | $l_{max}$ | 7 | 7 | 9 | 9 | 9 | 7 | 9 | 7 |
| | $\alpha$ | 2 | 1 | 1 | 1 | 1 | 20 | 1 | 1 |
| | $p_{min}$ | $10^{-6}$ | $10^{-10}$ | $10^{-6}$ | $10^{-7}$ | $10^{-6}$ | $10^{-5}$ | $10^{-7}$ | $10^{-10}$ |
| | $\lambda$ | 0.05 | 0.20 | 0.10 | 0.20 | 0.10 | 0.20 | 0.05 | 0.20 |
| | | Showing Condition | | | | | | | |
| | | ooo | oox | oxo | oxx | xoo | xox | xxo | xxx |
| Standard Planning Model | $\lambda$ | 0.08 | 0.10 | 0.02 | 0.02 | 0.30 | 0.09 | 0.02 | 0.02 |
| Pedagogical Model | $l_{max}$ | 7 | 9 | 9 | 9 | 11 | 9 | 9 | 7 |
| | $\alpha$ | 1 | 10 | 20 | 1 | 1 | 5 | 1 | 1 |
| | $p_{min}$ | $10^{-10}$ | $10^{-7}$ | $10^{-7}$ | $10^{-7}$ | $10^{-10}$ | $10^{-5}$ | $10^{-7}$ | $10^{-10}$ |
| | $\lambda$ | 0.20 | 0.05 | 0.05 | 0.20 | 0.20 | 0.05 | 0.05 | 0.20 |

Note: The codes for the reward functions refer to which tiles were safe (o) and which were dangerous (x) with the ordering <orange, purple, cyan>.

## Footnotes

[1] For $a, b, c, d > 0$ if $a \geq b$ and $c > d$, then: