[Reviews · NeurIPS 2016]

Reviewer 1

Summary

This well-written paper presents a new approach to learning from demonstrations, introducing an algorithm for picking the best trajectories for demonstrations as well as learning reward functions from such teaching trajectories. They show that the teaching algorithms qualitatively matches human behaviour and the algorithms lead to better learning from demonstrations in general.

Qualitative Assessment

The paper provides a new and interesting approach to the problem of learning from demonstration. The paper effectively tackles both sides of the question laid out, developing algorithms both for better teaching and for better learning (assuming a teacher). The algorithms are validated in a pair of experiments with humans and through simulations. Below are some suggestions for improvement: 1. I found it hard to track exactly which model was being used in which experiment/simulation. For example, I don’t see the pedagogical IRL model being used in any of the experiments. As far as I can tell, the first experiment showed the difference between the standard planning model and the pedagogical model. The second experiment also showed the difference and then used standard IRL to infer reward functions from both generated trajectories and human demonstrators (do or show). 2. The general picture of the approach is quite clear, but the details of the algorithms from the equations and the algorithm table were not sufficiently clear. Algorithm 1 is not really discussed in the text, nor unpacked, and the equation have multiple undefined terms, whose functionality was not obvious (esp Eq 2 and the subsequent unnumbered one). What are d and h in Eq 2? How does the parameter alpha change things? Is it the same parameter in the two equations? 3. For all the stats, means could use a measure of variance (e.g., confidence interval or SEM). Why are some of the t-tests one-sided and others two-sided? There does not seem to be an obvious logic. Why are the degrees of freedom fractional and different from one test to the next? 4. The design of experiment 2 was hard to discern from the figure alone. I think the experiment display was the coloured blocks, and the hidden reward function was the first column, but it took several reads to pull that out. A couple more basic introductory sentences would help. Was the reward 10 points or 5 points (seems to be stated different in 2 places)? 5. The general discussion was very thing and made surprisingly little effort to establish the contribution of this paper to the larger literature on teaching by demonstration, inverse reinforcement learning, or even wider impact on pedagogy at large.

Confidence in this Review

2-Confident (read it all; understood it all reasonably well)


Reviewer 2

Summary

A model is presented for how a sequential decision maker could behave if the task is to inform a learner about the underlying reward function. In two simple navigation experiments the model is compared to human behavior.

Qualitative Assessment

I find it very interesting to model how a rational agent or human could behave when teaching. But the paper does not fully succeed to convince me that the proposed model is a good one. This is mostly because the comparison with human data is rather qualitative and focuses only on few aspects. 1. For example figure 1 gives the impression that the model and humans act very similarly in the standard planning (doing) condition. Isn't this just an effect of only showing the 2 most probable trajectories of the model? I guess, the total probability of a (swerving) trajectory going through the diagonal square between start and goal is more likely than the total probability of the shown trajectories, i.e. P(up-right-up-right) + P(up-right-right-up) + P(right-up-right-up) + P(right-up-up-right) > P(up-up-right-right) + P(right-right-up-up). I don't have a good intuition for the model distribution in the showing condition, but I wouldn't be surprised to find similar discrepancies. A comparison of the trajectory probabilities in a P-P-plot could be revealing. 2. It feels unsatisfactory that a prior favoring short sequences needs to be used to match the data in in experiment 2 (line 218-220). In fact, I think this is another weakness of the model. How would the values in table 1 change without this extra assumption? 3. I didn't find all parameter values. What are the model parameters for task 1? What lambda was chosen for the Boltzmann policy. But more importantly: How were the parameters chosen? Maximum likelihood estimates? 4. An answer to this point may be beyond the scope of this work, but it may be interesting to think about it. It is mentioned (lines 104-106) that "the examples [...] should maximally disambiguate the concept being taught from other possible concepts". How is disambiguation measured? How can disambiguation be maximized? Could there be an information theoretic approach to these questions? Something like: the teacher chooses samples that maximally reduce the entropy of the assumed posterior of the student. Does the proposed model do that? Minor points: • line 88: The optimal policy is deterministic. Hence I'm a bit confused by "the stochastic optimal policy". Is above defined "the Boltzmann policy" meant? • What is d and h in equation 2? • line 108: "to calculate this ..." What is meant by "this"? • Algorithm 1: Require should also include epsilon. Does line 1 initialize the set of policies to an empty set? Are the policies in line 4 added to this set? Does calculateActionValues return the Q* defined in line 75? What is M in line 6? How should p_min be chosen? Why is p_min needed anyway? • Experiment 2: Is the reward 10 points (line 178) or 5 points (line 196)? • Experiment 2: Is 0A the condition where all tiles are dangerous? Why are the likelihoods so much larger for 0A? Is it reasonable to average over likelihoods that differ by more than an order of magnitude (0A vs 2A-C)? • Text and formulas should be carefully checked for typos (e.g. line 10 in Algorithm 1: delta > epsilon; line 217: 1^-6;)

Confidence in this Review

2-Confident (read it all; understood it all reasonably well)


Reviewer 3

Summary

This paper addresses the subject of teaching by demonstration. In particular, it asks the following question: When teaching from demonstration, is there a better way to form trajectories than following the optimal policy? In addressing this question, the authors combine two earlier strands of research in the literature, inverse reinforcement learning and Bayesian pedagogy, to introduce a new model called pedagogical inverse reinforcement learning. They perform two experiments with human subjects on two simple, deterministic gridworld problems, using a tabular and a feature-based reward function. The authors observe that people differ in the trajectories they produce when performing the task for themselves (to maximize return) and when demonstrating the task to others (so that they can maximize return as well).

Qualitative Assessment

The subject of this paper (teaching by demonstration) is important and relevant for NIPS. The primary questions that the authors ask are the following: -- How might showing how to do a task differ from merely doing it? -- How can inverse RL algorithms benefit from intentional teaching? These questions are novel and this line of research can potentially have high impact. The model introduced by the authors is a straightforward combination of two pieces of earlier work on inverse reinforcement learning and on Bayesian pedagogy. From this model, the authors make certain predictions, then conduct two experiments with human subjects to evaluate whether these predictions are observed in behavioral data. In the paragraph that starts on line 124, the authors state that the pedagogical model makes two types of qualitative predictions. The first prediction is that certain trajectories show "clearer" paths than others. The second prediction is that certain "looping" behaviors will be observed. I have a number of concerns regarding these qualitative predictions. My first concern is that these predictions are stated only verbally, not mathematically, and it is not entirely clear what the authors mean by them. For example, given two paths to the goal, how can we determine which one is a "clearer" path? Does the answer generalize to stochastic domains? Second, I do not follow how these predictions follow from the model in section 2.3. It would be useful if the authors can show how their model leads to these predictions (for example, can they derive them analytically?). It would also improve the paper to state the predictions clearly with no ambiguity. Third, it is not clear how the two types of predicted behaviors are actually useful for learning from demonstrations. For example, given that the goal states are distinct and far from each other, why is it desirable that the paths do not intersect? The analysis of the two experiments are not as rigorous as one would like, and some experimental details are left out, which makes it difficult to interpret the results. Detailed comments below. EXPERIMENT 1 1. What was the role of the learning block? From the text, I understand that in both the "Do" and "Show" conditions, the subjects were explicitly shown where the reward is. And if there were any participants learning from demonstration (the text is not clear on this), these participants observed deterministic trajectories generated by people who knew where the reward is, which means that they could observe the reward location indirectly, without having to learn. 2. Was the length of the trajectory important? That is, were shorter trajectories to the goal state more rewarding than longer trajectories? How was that information conveyed to the subjects? More generally, what exactly was the underlying reward function? 3. On line 148, the authors write that the participants in the "show" condition were more likely to choose paths that helped disambiguate their goal as compared to those in the "do" condition. As noted above, it is not clear why this type of disambiguation is useful. Was it indeed useful to the participants who learned by demonstration (if they did exist; see comment 1 above)? That is, was there a difference in the success rates of the participants who learned from demonstration when presented with the different trajectories? 4. What does M denote on line 151 (also used later)? I assume it is mean of something, but of what? 5. I do not follow the sentence that starts on line 152: "For trials where..." How does the model predict marginal significance? 6. Based on lines 145-146, I conclude that each participant completed 9 trials and that these nine trials used the 3 tasks shown in Figure 1, repeating each task 3 times. Is this correct? If yes, what is the rest of the data (Figure 1 shows only 3 trials)? If not, the paper should describe all 9 trials. 7. Why does the model predict loping behavior when % is the goal? (as stated in Figure 1 caption) 8. I do not see an analysis of the % goal. The difference between the "do" and "show" conditions in this condition is much smaller than in the & condition. Why is that? Could there be an order effect? 9. Figure 1 left-hand column is showing trajectories from only 2 of the standard planning algorithms. This is misleading because these two trajectories are not special in any way. Given that the domain is deterministic, a number of other trajectories exist that collect exactly the same return as the two trajectories that are shown. On a related note, on the second column from the left, only the two model trajectories with the highest probabilities are presented --- it would be much more informative to see a larger number of trajectories, along with the actual probabilities assigned to them by the model. This information would show, for instance, whether there is a third trajectory with identical probability or whether there is a large gap in probability between the first and the second trajectory. 10. In section 3.3, it would be useful to see a high-level discussion of the overlap between model predictions and human behavior. EXPERIMENT 2 11. There is an inconsistency in the terminal reward given. Is it 5 (line 196) or 10 (line 178)? 12. It would be nice to see a more direct link from the pedagogical model to the specific types of trajectories described in lines 187-190. (I have made similar comments earlier regarding Experiment 1.) 13. Columns 2 and 3 in Figure 2: See comment 9 above. 14. Line 211: "potentially suboptimal paths": Longer paths on safe tiles are *not* suboptimal; they collect exactly the same return as the shorter paths in safe tiles (unless it was communicated to the participants in some way that shorter trajectories are better, but I see no indication of this) 15. Assertions in lines 212-214 should be backed up with a quantitative analysis. 16. Line 217: The choice of alpha=1.5 should be discussed. In addition, it would be useful to have an analysis of the sensitivity of the results to the value of this parameter. 17. Lines 217-221: This introduces a difficulty in interpreting the results and should be discussed in some detail. In addition, using gamma=1 but adding a small negative reward for each action would place the two approaches at equal footing, is this not correct? 18. Most numbers in Table 1 are very small (except for the 0A condition), indicating that the observed trajectories are not a good fit to either model. Furthermore, the overall difference between "show" and "do" conditions is very small (as deduced from the last column in Table 1: about 0.02). Consequently, I do not think that the statements in lines 222-228 are a fair and accurate description of the results in Table 1. 19. Lines 237-239: Text is inconsistent with the numbers shown in parenthesis. OTHER -- Mathematical symbols in Equation 2 should be defined. -- Algorithm 1, line 5: set pi (initialized on line 1) is still empty so this line would lead to j being assigned to the empty set.

Confidence in this Review

2-Confident (read it all; understood it all reasonably well)


Reviewer 4

Summary

This is a nice paper that expands on a bayesian model of teaching introduced in a 2014 cognitive psychology paper. The main innovation here is to distinguish between doing and teaching by demonstration (showing). In the original paper, the model is used to predict examples used for teaching purposes in three different settings. In this paper, the model is used to compare the performance of humans and predictions of either an MDP solver or a pedagogical model when the humans are doing a task versus trying to show another person what the goal is in a simple grid world. The model results predict the human behavior fairly well. In a second experiment, there are features involved in the grid world that may signal penalties (colored tiles), and the human demonstrations perform suboptimal paths to the goal that are nevertheless informative about which tiles are dangerous and which are safe. Again, the MDP solver predicts the human behavior in doing the task better than when demonstrating it, and the pedagogical model predicts the human behavior when demonstrating the task better than when they are simply doing it. Finally, they show that a state of the art inverse reinforcement learning model (MLIRL) learns a more accurate representation of the environment (which tiles are safe vs. dangerous) from either the human or the model demonstrations than from the human or model simply doing the task.

Qualitative Assessment

Specific comments/typos/wording: Citations not in reference list: Lyons et al. 2007, Abbeel et al., 2007. In references, Babes -> Babes-Vroman
 Algorithm 1 is not apparently referenced or explained in the text. It should be pushed to supplementary material with a detailed explanation. I note that as written, lines 11-14 will never be reached. You set delta = infinity, and then the while loop runs as long as delta is less than epsilon. Furthermore, on line 1 you set PI to the empty set, and never add to it. Thus line 5 would lead to a null j, since you calculate j such that there exists a pi element of PI such that….Thus, j will be null. I assume you want to add pi to PI after line 4. Those are just the obvious bugs; without more explanation, this algorithm is relatively impenetrable. Equation 2 needs more explanation. As written, it is unclear why it is correct. Basically, it says that x is proportional to (xy)^alpha, which is nonsensical unless alpha=1. line 142: won-> win line 148: As -> that, as line 152-154 - “as predicted by the model” appears twice. Also, you are describing the statistics for “&”, which you just described in the previous sentence. One of these much be “%.” The model does not account for the swervng behavior for the “#” goal. How would you change it to account for this behavior? Table 1: you haven’t told us what the “0A” or “1B” conditions are, and these are the conditions most strongly predicted by the model. It would behoove you to exhibit these, as well as to provide some explanation as to why these are so strongly predicted, especially 0A. I understand that it is the relative predictions that matter, but these numbers are so wildly different that it would be good to know why. Lines 237-240: the prediction is opposite of what you say it is (Show: M=.82, Do: M=0.87). Again, here, you refer to reward functions that aren’t explained.

Confidence in this Review

2-Confident (read it all; understood it all reasonably well)


Reviewer 5

Summary

In inverse reinforcement learning (IRL), an agent learns a reward function by looking at the action of another agent. Based on the fact that a human teacher modifies their behavior when demonstrating a task to a student in order to facilitate learning, the paper looks at two different strategies for the demonstrating system to teach the observing one, one called doing (where the teaching system behaves the same way with or without a student) and showing (where the teaching system modifies its behavior in order to facilitate learning by the students). A formal framework based on a Bayesian analysis is developed in the first part of the paper and two experiments with human participants try to give it empirical supports. The two experiments show that, in a teaching situation, human participants change their behavior in a way that might be consistent with the formal framework and that eases learning of the value function by an agent using IRL.

Qualitative Assessment

The paper tackles a very interesting topic as a vast literature in psychology indeed shows that human do change their behavior depending what they might to communicate to another person, notably in a teaching situation. I do not have the mathematical skills to fully appreciate the framework the authors are proposing but as far as I can tell, it looks promising. The problem is how to test it is this is supposed not only to be a method to boost IRL but also a model of human behavior, which seems to be the goal of the authors. I am not totally convinced by Experiment 1 notably as the statistics do not look appropriate to me. First, it is not clear what the dependent variable is. Intuitively, I would say it is the proportion of trials a participant used a given strategy but sentence like "more people took the outside route in the show condition" got me confused: It either does not correspond to the way the analysis were run, or I am missing something. In both cases, more details are necessary. Moreover, especially in those days where p-hacking is a concern, the lowest number of statistical analysis should be run. In this case, a 2x2 ANOVA of conditions (doing vs. showing) and goal (#,%,&) would have been the appropriate analysis, eventually followed by the t-tests the authors did in the paper. I am not sure this analysis would bring any main effect of condition or an interaction of the two factors. Finally, I am not sure I understand why no difference between the conditions is predicted for the # target. More information should be given about this. Overall, I am not sure experiment 1 is worth putting the paper as experiment 2 basically makes the same point but with a much more interesting task and with a better statistical argument. Yet, I have two remarks. First, the reward function maps shown in Figure 2 are not the easiest way to realize whether or not showing has lead to a better IRL than doing. Maybe showing the amount of reward collected by an agent following those maps (maybe relative to an agent following the optimal strategy) would allow to see that more immediately. Second, the authors conclude that in the showing condition, the action of the participants are better explained by the pedagogical model in the showing condition. It might be could it just be because the behavior becomes more randomlike (increased entropy) in the teaching condition? Would any model leading to a more randomlike behavior would account as well for the data (like for instance, a straighforward RL algorithm using a softmax action selection rule whose inverse temperature parameter is increased in the showing condition) or is this predictive advantage specific to the pedagogical model? This might provide a better null hypothesis. Also, it is not clear how the authors chose free parameters for the pedagogical model (nor what some of these parameters like lmax pr pmin: they are not mentioned elsewhere in the paper).

Confidence in this Review

2-Confident (read it all; understood it all reasonably well)


Reviewer 6

Summary

PaperID 1506, Showing versus doing: Teaching by demonstration, proposes a novel algorithm and computational framework for sequence learning by building on inverse reinforcement learning and Bayesian pedagogy. The resultant model, termed Pedagogical Inverse Reinforcement Learning (PIRL), distinguishes itself from previous models by selecting trajectories in a given state space that "show" how to perform a task rather than solve the problem. The PIRL model predicts several differences from the standard planning model encapsulated by conventional Markov Decision Processes. 1. For a given scenario with multiple end-goals, PIRL will choose the sequence of actions that best separates a particular goal from other goals. 2. PRIL will choose suboptimal sequences compared to an agent seeking the optimum solution to achieve a goal in a sequence of steps. The choice of suboptimal sequences are intended to signal the reward value of state features. An algorithm is presented in the paper that shows how a particular policy is chosen and built upon based on action values and specified trajectory for a given observer. The behavior of the PIRL algorithm are tested against human behavior collected in mechanical turk experiments. In experiment 1, the author shows that people modify their behavior to signal information about goals when they are tasked to demonstrate to others. This experiment is performed in a simple grid game in which an agent is attempting to reach one of three possible goals for each session of the game. Three goals are defined in particular positions on the grid such that goal-directed sequences with both overlapping and non-overlapping states existed. The PIRL algorithm predicts that a demonstrator seeking to teach others where the goal was would choose the goal-directed sequence with the least overlap with other possible sequences that led to other goals. Overall, human actors exhibited this behavior when instructed to demonstrate the task to others but not when asked to just accomplish the task. This suggests that there is a difference in human behavior in the context of teaching others by doing versus just accomplishing a task. Experiment 2 builds up on experiment 1 by seeking to demonstrate how showing how to do a task (instead of just doing the task) could make a difference on how well the task is learned. This time a colored-tile grid game was used to study feature-based reward functions. Tiles could be different colors which correspond to positive or negative rewards. White tiles are always considered "safe." Where as colored tiles could be either safe or incur a reward penalty. A yellow tile signified the end-point goal. In this task, PIRL predicts pedagogical behavior that is suboptimal for accomplishing the task. An agent concerned merely with an optimal solution will generally take the shortest path between the starting point and the goal. These predictions were evaluated via a human behavioral experiment using Mechanical Turk. Humans that were instructed to perform the task, generally did so in a manner that was optimal for solving the task. However, humans that were instructed to demonstrate the task to others, undertook suboptimal (longer) paths to demonstrate the features of the individual tiles (i.e. that some colors may be bad! so stick with one set of features no matter the path, or loop across a variety of safe tiles). The information entropy was calculated on the human behavioral data to quantify the redundancy (or variety) of visited tiles. Average entropy was higher in the show condition, suggesting that people will visit a greater variety of safe tiles when showing a task. In addition to evaluating the predictions of their algorithm by validating it's performance with human behavioral data, the authors also extend their results to training standard inverse reinforcement learning algorithms. These models do not assume that the demonstrator is trying to teach, however feeding PIRL trajectories to these models was predicted to result in better learning. This was tested using a Maximum Likelihood Inverse Reinforcement Learning Algorithm (MLIRL). Overall, the MLIRL was able to better represent the associated reward function across all the tiles for the "show" compared to the "do" condition.

Qualitative Assessment

Overall, this paper is a well written piece that provides novel contributions to the field of reinforcement learning. Firstly, the paper does an excellent job demonstrating that human behavior changes depending on whether an individual seeks to accomplish or demonstrate a task to others. Secondly, the authors present a novel algorithm that very closely predicts human behavior for both doing and showing how to accomplish a task. Lastly, the proposed algorithm is shown to further improve existing reinforcement learning models. The results of this paper are very exciting and refreshing to read about in a field littered with advancements based solely on mathematical acrobatics. I deeply enjoyed the bridging between human behavior and computational models for artificial intelligent behavior. The only criticism I can give on this paper is that equation one is not adequately explained for those that are outside of the field. The terms are not defined and the importance of the Markov Decision Process to the rest of the paper is not readily apparent. Also the last four columns of Figure 2 are not exceptionally clear. It is demonstrated that the algorithm learns more about the tiles and their associated rewards (via maximum-likelihood) but the figure is hard to interpret without a color bar (i.e. does white mean anything? is black indicative of a low likelihood?). I would greatly appreciate learning more about this algorithm in person during an oral presentation. I would recommend a higher rating for this paper, however I do not have the appropriate training to truly gauge the importance of the proposed model. Also, further validation on other reinforcement algorithms would be desirable.

Confidence in this Review

2-Confident (read it all; understood it all reasonably well)